# Vaccination with Plasmids Encoding the Fusion Proteins D-S1, D-S1N and O-SN from SARS-CoV-2 Induces an Effective Humoral and Cellular Immune Response in Mice

**DOI:** 10.3390/vaccines13020134

**Published:** 2025-01-28

**Authors:** Noe Juvenal Mendoza-Ramírez, Julio García-Cordero, Gabriela Hernández-Galicia, Nicole Justine Moreno-Licona, Jesus Hernandez, Carlos Cabello-Gutierrez, Joaquín Alejandro Zúñiga-Ramos, Edgar Morales-Rios, Sonia Mayra Pérez-Tapia, Vianney Ortiz-Navarrete, Martha Espinosa-Cantellano, David Andrés Fernández-Benavides, Leticia Cedillo-Barrón

**Affiliations:** 1Departamento de Biomedicina Molecular CINVESTAV, Av. IPN # 2508 Col, San Pedro Zacatenco 07360, Mexico; noe.mendoza@cinvestav.mx (N.J.M.-R.); jugarcia@cinvestav.mx (J.G.-C.); gabriela.hernandez@cinvestav.mx (G.H.-G.);; 2Departamento de Bioquímica Cinvestav, Av. IPN # 2508 Col, San Pedro Zacatenco 07360, Mexico; nicole.moreno@cinvestav.mx (N.J.M.-L.); edgar.morales@cinvestav.mx (E.M.-R.); 3Laboratorio de Inmunología, Centro de Investigación en Alimentación y Desarrollo A. C (CIAD) Carretera a la Victoria km 0.6, Hermosillo Sonora 83304, Mexico; 4Departamento de Investigación en Virología y Micología, Instituto Nacional de Enfermedades Respiratorias Ismael Cosío Villegas (INER), Calzada de Tlalpan 4502, Belisario Domínguez, Tlalpan 14080, Mexico; carloscg@iner.gob.mx (C.C.-G.); joaquin.zuniga@iner.gob.mx (J.A.Z.-R.); 5Escuela de Medicina y Ciencias de la Salud, Tecnológico de Monterrey, Monterrey 64710, Mexico; 6Unidad de Desarrollo e Investigación en Bioterapéuticos (UDIBI), Escuela Nacional de Ciencias Biológicas, Instituto Politécnico Nacional, México City 11340, Mexico; mayra.perez@udibi.com.mx; 7Departamento de Infectómica y Patogénesis Molecular, CINVESTAV, Av. IPN # 2508 Col, San Pedro Zacatenco 07360, Mexico; mespinosac@cinvestav.mx; 8Centro de Ingeniería y Desarrollo Industrial (CIDESI), Av. Playa Pie de la Cuesta No. 702, Desarrollo San Pablo, Querétaro 76125, Mexico; david.fernandez@cidesi.edu.mx

**Keywords:** vaccine, nucleocapsid, plasmid DNA, fusion proteins, DNA immunization

## Abstract

Background: Next-generation vaccines against coronavirus disease 2019 (COVID-19) focus on inducing a long-lasting immune response against severe acute respiratory syndrome coronavirus-2 (SARS-CoV-2) and its emerging variants. To achieve this, antigens other than spike proteins have been proposed, and different platforms have been evaluated. Nucleic acid-based vaccines are fundamental for this process. Preclinical data have shown that the SARS-CoV-2 nucleocapsid protein induces a protective cellular immune response, and when combined with the spike protein, the resulting humoral and cellular immune responses are effective against some SARS-CoV-2 variants. Methods: We designed a DNA vaccine against the spike and nucleocapsid proteins of SARS-CoV-2 to generate fusion proteins based on the Delta and Omicron B.5 strains. The most immunogenic regions of the spike and nucleocapsid proteins of the Delta and Omicron B strains were selected using bioinformatics. The nucleotide sequences were cloned into pcDNA3.1, and named pcDNA3.1/D-S1, pcDNA3.1/D-S1N, and pcDNA3.1/O-SN. The immunogenicity of the generated fusion proteins was evaluated by analyzing the humoral and cellular responses elicited after the immunization of BALB/c mice. Results: DNA immunization induced antibody production, neutralization activity, and IFN-γ production. The inclusion of the nucleocapsid regions in the plasmid greatly enhanced the immune response. Moreover, cross-reactions with the variants of interest were confirmed. Conclusions: Plasmids-encoding fusion proteins combining the most immunogenic regions of the spike and nucleocapsid proteins present a promising strategy for designing new and effective vaccines against SARS-CoV-2.

## 1. Introduction

Coronavirus disease 2019 (COVID-19) is caused by the severe acute respiratory syndrome coronavirus-2 (SARS-CoV-2). The virus has spread worldwide [1] and has been responsible for over 776,841,264 deaths since its emergence in December 2019 [2]. The SARS-CoV-2 RNA genome encodes four structural proteins: spike (S), nucleocapsid (N), membrane (M), and envelope (E) [3]. Spike proteins are highly glycosylated trimeric proteins that mediate viral entry into cells by binding to angiotensin-converting enzyme 2 receptors (ACE2) [4]. Spike proteins contain a polybasic site that yields subunits 1 (S1) and 2 (S2) after protease cleavage. S1 contains the receptor-binding domain (RBD), which is involved in the interaction with ACE2, whereas S2 contains the fusion peptide involved in the fusion of the viral membrane with the host cell membrane. Most neutralizing antibodies (nAbs) reported to date are directed against the RBD [5], although other targets of the S protein include the S1 N-terminal domain (NTD), protease cleavage enzyme, S2 stem helix, and the fusion peptide [6].

Most vaccine platforms developed during the SARS-CoV-2 pandemic are directed against spike proteins [7]. Since then, numerous mutations have emerged in the SARS-CoV-2 genome, particularly in the spike protein-encoding sequence, resulting in the emergence of several variants [8]. Consequently, the effectiveness and neutralization activity of the main protective antigens against variants of concern (VOCs) could be variable [9]. Therefore, next-generation COVID-19 vaccines must focus on inducing strong and long-lasting immunity against the original Wuhan SARS-CoV-2 strain and its variants [10]. The development of these new vaccines explores the use of antigens other than spike proteins, and different platforms and new formulations have been evaluated in recent research [10]. Structural proteins of SARS-CoV-2, such as the nucleocapsid protein (N), have been evaluated in preclinical studies [11]. This protein is highly conserved among coronaviruses and their variants and induces humoral and long-lasting cellular immune responses [11,12,13].

Nucleocapsids contain two RNA-binding domains (NTD and CTD) and a central Ser/Arg-rich flexible linker region (LKR) which is involved in viral pathogenesis, replication, and RNA packaging [14]. Preclinical studies have shown that immunization with nucleocapsid proteins induces the production of IgM, IgG, and IgA antibodies (Abs). Additionally, IgG against the N protein activates natural killer cells and induces Ab-dependent cell-mediated cytotoxicity (ADCC) [15,16,17]. Immunization with N also induces activation of CD4^+^ and CD8^+^ T cells and production of cytokines such as IFN-γ, TNF-α, IL-2, IL-4, and Granzyme B [18,19,20]. Additionally, immunization with a combination of spike and nucleocapsid proteins has been shown to induce neutralization and cellular activity. Several studies have reported enhanced humoral and cellular immune responses when these two antigens are combined, and such responses are effective against different VOCs [21,22,23,24].

As previously mentioned, next-generation COVID-19 vaccines explore the use of different platforms to present antigens. Advances in nucleic acid-based vaccines have marked a new era in the field of vaccine development. Pharmaceutical companies such as Moderna and Pfizer have developed and administered mRNA-based vaccines worldwide, and their efficacy, safety, and security have been confirmed [25]. DNA-based nucleic acid vaccine technology has been developed for other viruses such as VPH, HIV, and Ebola. In contrast to mRNA vaccines, DNA vaccines are relatively cheap, and their production is rapidly scalable; they are highly stable and can be stored for long periods [26]. Therefore, several DNA vaccines against SARS-CoV-2 are being tested in the preclinical phase and clinical trials 1 or 2 [27].

In this study, we use bioinformatic approaches to design plasmids that encode fusion proteins with the most immunogenic regions of both the spike and nucleocapsid proteins of SARS-CoV-2, based on Delta and Omicron B.5 strain sequences. The sequences were cloned into pcDNA3.1, and named pcDNA3.1/D-S1, pcDNA3.1/D-S1N, and pcDNA3.1/O-SN. We evaluated the immune responses elicited after immunization with these plasmids.

## 2. Materials and Methods

### 2.1. Design of Fusion Proteins

Enriched regions with the most immunogenic peptides in the S and N sequence were selected for the design of fusion proteins. The selection of regions in Spike protein was focused on the capacity to induce an antibody response. Most spike nAbs that have been studied so far bind to the RBD; therefore, for this work, we included the whole RBD in the chimeric protein. Some nAbs have been described in the N-terminus of the S protein, so we selected epitopes present in this region. GSGSGS linkers were added between selected regions from the spike protein. The selection of regions in nucleocapsid protein was focused on the capacity to induce T cellular response. To include as many epitopes as possible, we selected three regions of N. We began with the Linker region and then in the C terminus, and AYY linkers were added between these regions. Other proteins were designed with spike regions alone. The first fusion protein was named D-S1 and it had the two regions selected from the spike protein (Figure 1a). The second fusion protein was named D-S1N, and it contained the same two regions from S plus the two regions selected from the nucleocapsid protein. Finally, a third fusion protein was designed with regions of full spike and nucleocapsid protein; this one was named O-SN. We added an 8-histidine tag in the C-terminus in all fusion proteins.

### 2.2. Bioinformatic Analysis

The selected regions were connected sequentially by linkers, and the structure of Fusion proteins was predicted by the AlphaFold server. Parameters such as “de novo” prediction and minimization of energy were applied. Visualization and analysis of this structure were performed using Discovery Studio. The physicochemical properties of the fusion proteins, including their molecular weight, theoretical isoelectric point, and hydrophobicity, were predicted in the ProtParam server (https://web.expasy.org/protparam/, accessed on 9 December 2024). Prediction of N-glycosylation sites was performed by NetNGlyc Server (https://services.healthtech.dtu.dk/services/NetNGlyc-1.0/, accessed on 9 December 2024). Prediction of antigenicity of the fusion protein was performed in Vaxijen (https://www.ddg-pharmfac.net/vaxijen/VaxiJen/VaxiJen.html, accessed on 9 December 2024). Fusion proteins were analyzed via the AllergenFP V.1.0 (https://ddg-pharmfac.net/AllergenFP/, accessed on 9 December 2024) server to confirm the allergen properties. To predict the fusion protein’s ability to activate an innate immune response, molecular docking with some TLRs was performed. Structures of TLR3 (PDB: 1ziw) and TLR4 (PDB: 3vq2) were obtained from the protein databank. All structures were previously prepared in Chimera Software V1.15. After that, molecular docking was performed in ClusPro (https://cluspro.bu.edu/login.php, accessed on 9 December 2024) server and the models obtained were analyzed in BIOVIA Discovery Studio Visualizer V.17.2.0.16349.

### 2.3. Gene Syntheses and Plasmid Construction

Sequence coding for amino acids selected in D-S1 and D-S1N fusion protein was based on the Delta strain of SARS-CoV-2 (OK091006.1). The sequence coding for amino acids selected in O-S1 was based on the Omicron strain BA.5.1 (EPI_ISL_13821991). Additionally, a histidine-tag (8xHis)-encoding sequence tag was added at the C terminus. All the synthesized nucleotides were cloned into the mammalian expression vector pcDNA3.1 (Invitrogen, Carlsbad, CA, USA) and optimized for expression in eukaryotic cells. The plasmids were named pcDNA3.1/D-S, pcDNA3.1/D-S1N, and pcDNA3.1/O-SN. Their insertion into pcDNA3.1 was confirmed via enzyme analysis and by SANGER sequencing.

### 2.4. Expression of Fusion Proteins

The plasmids pcDNA3.1/D-S1, pcDNA3.1/D-S1N, and pcDNA3.1/O-SN were transiently transfected into Vero (Bellco, Vineland, NJ, USA) and Expi293 cells (Thermo Fisher, Waltham, MA, USA) and their expression was evaluated by Western blot and immunofluorescence.

### 2.5. Western Blotting

The supernatant of Expi293-transfected cells was analyzed in SDS-PAGE 8% and then electro-transferred to polyvinylidene difluoride membranes (Hybond ECL; GE Healthcare, Little Chalfont, UK). Membranes were blocked with semi-skimmed Milk/PBS-Tween 0.1% and then incubated with the primary antibody, anti-His-HRP, overnight at 4 °C. After washing with PBS-Tween 0.1%, the membranes were visualized with a Western lightning-enhanced chemiluminescence reagent (Pearce, Rockford, IL, USA).

### 2.6. Immunofluorescence

To evaluate intracellular expression of fusion proteins, Vero cells were seeded on glass coverslips (6 × 10^4^) and transfected with pcDNA3.1/D-S1, pcDNA3.1/D-S1N, pcDNA3.1/O-SN, or parental plasmid. Twenty hours post-transfection, Brefeldin A (9972S, Cell Signaling Technology^®^, Beverly, MA, USA), was added and four hours after this, the cells were fixed with 4% paraformaldehyde (Sigma-Aldrich, St. Louis, MO, USA) and washed with PBS 1X. To evaluate the recognition of SARS-CoV-2, using serum samples, Vero cells were infected for 24 h in a Biosafety Laboratory Level 3 (BSL-3, Instituto Nacional de Enfermedades Respiratorias, Mexico City, Mexico), as previously described [22]. Cells were fixed, permeabilizated, and incubated with serum samples from patients with COVID-19 (1:200 dilution), pre-pandemic serum, mouse polyclonal antibodies against spike and nucleocapsid proteins of SARS-CoV-2 (1:200 dilution), or serum samples from immunized mice. For human sera, human IgG-FITC (81-7111, Zymax) was added and incubated for 1 h and washed 5 times. For mouse polyclonal antibodies, mouse IgG-Cy3 (A10521, Invitrogen, Carlsbad, CA, USA) and mouse IgG-Alexa488 (A11001, Invitrogen, Carlsbad, CA, USA) were added for 1 h and washed 5 times. Finally, nuclei were labeled with DAPI/Vectashield (H-1200, Vector Labs, Burlingame, CA, USA). Images were taken using a confocal microscope (Leica SP8).

### 2.7. Immunization

Female 6–8-week-old BALB/c mice were immunized 3 times (at days 0, 20, 40) by intramuscular injection. The first experiment was used to evaluate the optimal dose required to induce an immune response. Each mouse was vaccinated with 10, 25, or 40 μg of DNA: pcDNA3.1/D-S1, pcDNA3.1/D-S1N, or pcDNA3.1/O-SN. Mice vaccinated with pcDNA3.1/S1, pcDNA3.1/N, and pcDNA3.1 were used as controls. Serum was collected from the blood on days 0, 20, and 40 and stored at −20 °C. The second immunization schedule was performed in BALB/c mice that were immunized 3 times (at days 0, 20, 40) by intramuscular injection with 25 μg of DNA. Six groups (pcDNA3.1/D-S1, pcDNA3.1/D-S1N, pcDNA3.1/O.SN, pcDNA3.1/S1, pcDNA3.1/N and pcDNA3.1) of 10 mice were evaluated. Serum was collected from the blood on days 0, 20, and 40 and stored at −20 °C. Euthanasia was performed 80 days post-priming in half of the mice. Final bleeding was performed 180 days post-priming in the other half of the mice. The Animal Use Ethical Committee at CINVESTAV approved the protocols and procedures (number 02-11-16).

### 2.8. Enzyme-Linked Immunosorbent Assay

ELISAs were performed as previously reported [22]. For variants, microplates were coated overnight at 4 °C with 4 µg RBD proteins based in Alpha, Gamma, Delta, and Omicron subvariants diluted in PBS 1X pH 7.4. Briefly, microplates were blocked with PBS containing 5% (*w*/*v*) skim milk for 1 h. Serial sera dilution was performed in PBS containing 5% (*w*/*v*) skim milk, and the plates were incubated for 2 h. Secondary antibodies (goat anti-mouse IgM-HRP (G21040, Invitrogen, Carlsbad, CA, USA) and goat anti-mouse IgG-HRP (62-6820, Invitrogen, Carlsbad, CA, USA)) were added. For isotype determination, the secondary antibodies rabbit anti-mouse IgG1-HRP (61-0120, Invitrogen, Carlsbad, CA, USA) and rabbit anti-mouse IgG2a-HRP (61-0220, Invitrogen, Carlsbad, CA, USA) were added. The plates were incubated for 1 h at 37 °C. After washing, o-phenylenediamine (P9029; Sigma–Aldrich, St. Lou-is, MO, USA) and H_2_O_2_ were used. The reaction proceeded for 15 min and was later stopped with 2 N H_2_SO_4_. The optical density was measured at 450 nm using an ELISA Lektor (Thermo Scientific Multiskan FC, Waltham, MA, USA).

### 2.9. Surrogate Virus Neutralization Test

An in-house surrogate virus neutralization test (sVNT) was used to evaluate the percentage of neutralization, in accordance with previous reports [28]. Briefly, samples were diluted with PBS (1:10) in a final volume of 60 µL. Serial dilutions were used in sera evaluation (1:10, 1:20, 1:40, 1:80, 1:160, 1:320). RBD-HRP was added to the samples and incubated for 30 min at 37 °C. The RBD–HRP mixture was incubated in a plate with recombinant ACE-2 for 15 min at 37 °C. The plate was washed and TMB was added and incubated for 20 min. Finally, the reaction was stopped with H_2_SO_4,_ and the optical density was measured using a spectrophotometer (Thermo Scientific Multiskan FC) at a wavelength of 450 nm. To determine the neutralization percentage, the following formula was used: %N = (1 − (Sample OD/Negative control OD)) ∗ 100.

### 2.10. Determination of IFN-γ Production

Mice were euthanized 67 days after the first immunization, and splenocytes were isolated. Splenocytes were cultured at 2 × 10^5^ density in 200 µL RPMI medium and stimulated with recombinant S1, N, RBD Delta, RBD XB4.5 (2 μg/mL) or positive control CD3 (in vivo ready, 40-0032-U100) plus CD28 (in vivo ready, 40-0281-M001) antibodies for 18 h at 37 °C. During the last 4 h of culture, Brefeldin A (BioLegend, 420601, San Diego, CA, USA) was added to the cell culture. Cells were stained for viability according to the instructions for the reagent, which was Zombie Green™ (BioLegend, 423111). After that, the following mouse antibodies were used to label cell surface markers: CD3-FITC (Invitrogen, 11-0031-86, Carlsbad, CA, USA), CD4-PE (Invitrogen, 12-0042-85, Carlsbad, CA, USA), and CD8-PerCP/Cy5.5 (BioLegend, 100734, San Diego, CA, USA). For intracellular staining, cells were permeabilized with Perm-wash (BD Biosciences, 554723, Franklin Lakes, NJ, USA) and stained with IFN-γ-PB (BioLegend, 505818, San Diego, CA, USA). Finally, cells were fixed with PFA 2%. Flow cytometry was carried out using Syxmex and the data were analyzed using Flowjo v10.10.10 software.

### 2.11. Proliferation Test

Splenocytes were stained with Cell Trace™ Far Red (Thermo Fisher Scientific, C34564, Waltham, MA, USA) according to the manufacturer’s protocol. After that, cells were seeded at 2 × 10^5^ density in RPMI medium and stimulated with recombinant S1, N, RBD Delta, RBD XB4.5 (2 μg/mL) or positive control CD3 (in vivo ready^TM^, 40-0032-U100, San Diego, CA, USA) plus CD28 (in vivo ready^TM^, 40-0281-M001, San Diego, CA, USA) antibodies for 5 days at 37 °C. Cells were stained with CD3-APC-CY7 and fixed with PFA 2%. Flow cytometry was carried out using Sysmex and data were analyzed using Flowjo v10.10.10 software

### 2.12. Statistical Analyses

A nonparametric ANOVA test in Prism v.8 software (GraphPad Software, San Diego, CA, USA) was used for statistical analyses. Bars represent the mean ± SD. * *p* < 0.05, ** *p* < 0.01, *** *p* < 0.001. *p* ≤ 0.05 was considered statistically significant

## 3. Results

### 3.1. Fusion Proteins D-S1, D-S1N, and O-SN Are Immunogenic According to Bioinformatic Approaches

Different plasmids were designed to theoretically express the most immunogenic regions of spike and nucleocapsid proteins. The regions were analyzed using programs such as ABCpred, B-cell epitope prediction tools, and T-cell tools on the IEDB server.

Selected regions were joined using GSGSGS/AAY linkers, and a His-tag was added at the end of each sequence, as described in Section 2 (Figure 1a–c). The three proteins designed were D-S1, D-S1N, and O-S1N. The physicochemical parameters were determined as shown in Figure 1a,b. D-S1 has 366 amino acids, a molecular weight of 41 kDa, and a theoretical isoelectric point (pI) of 9.11 (Figure 1). The instability index (II) was computed to be 24.23; therefore, the protein was classified as stable [29]. The aliphatic index was 74.73, and the grand average hydropathicity (GRAVY) was −0.264 [30]; thus, the protein was deemed hydrophilic. The fusion protein, D-S1N, has 456 amino acids, a molecular weight of 50.84 kDa, and a theoretical pI of 9.3. Its instability, aliphatic, and GRAVY indices were 29.72, 70.07, and −0.326, respectively. The fusion protein O-S1N has 517 amino acids, a molecular weight of 57.15 kDa, and a theoretical pI of 9.5. Its instability, aliphatic indices, and GRAVY values were 37.33, 59.07, and −0.538, respectively. Glycosylation sites were predicted for the three proteins, and four glycosylation sites were found in all three proteins.

The online software AllerTOP 2.0 defined all fusion proteins as nonallergenic. The UniProtKB accession numbers of the nearest proteins were Q8TEP8 (D-S1 and D-S1N) and Q68CP9 (O-S1N), which were also probable nonallergens. VaxiJen 2.0 showed that the overall prediction values for D-S1, D-S1N, and O-SN were 0.60, 0.64, and 0.58, respectively; the virus was selected as the prediction model, and the threshold for this model was 0.4. Therefore, the allergenicity and antigenicity of D-S1 and D-S1N are considered to be within safe levels.

An effective innate immune response is important to induce a successful adaptive immune response. Therefore, we evaluated whether the designed fusion proteins could activate innate receptors such as toll-like receptors (TLRs; Appendix A). The energy values of the amino acids involved in the interactions and the types of bonds are described in Appendix A. Through in silico observations, we found that D-S1 could bind to TLR3 and TLR4 mainly through hydrogen bonds, with energy values of −1019 Kcal/mol and −1343.7 Kcal/mol, respectively. Both D-S1N and O-SN could also interact with TLR3 and TLR4, with energy values of −1054.1 Kcal/mol and −1368.1 Kcal/mol for D-S1N, respectively, and −1256 Kcal/mol and −1244.1 Kcal/mol for O-SN, respectively. This suggests a higher binding efficiency. Additionally, N amino acids improved the binding of the fusion protein to TLR3, and probably improved receptor activation (Appendix A).

### 3.2. Evaluation of Fusion Protein Expression in Eukaryotic Cells

To evaluate the expression of the fusion proteins, Vero cells were transfected with pcDNA3.1/D-S1, pcDNA3.1/D-S1N, pcDNA3.1/O-SN, or parental pcDNA3.1, and at 24 h post-transfection, the cells were analyzed by immunofluorescence (Figure 2a). Using antibodies against S1, positive signals (green) were observed in cells transfected with pcDNA3.1/D-S1, pcDNA3.1/D-S1N, and pcDNA3.1/O-SN. We also used specific antibodies against the nucleocapsid and observed positive signals (red) only in cells transfected with pcDNA3.1/D-S1N or pcDNA3.1/O-SN (Figure 2a). To evaluate the secretion of the fusion protein, Expi293 cells were transfected with pcDNA3.1/D-S1, pcDNA3.1/D-S1N, or pcDNA3.1/O-SN, and the supernatants were assessed five days post-transfection by Western blot (Figure 2b). We used an antihistidine antibody and observed bands at 60, 80, and 90 kDa for D-S1, D-S1N, and O-SN, respectively.

Additionally, we confirmed the identity of the constructs when the cells were stained with sera from convalescent patients with COVID-19 and pre-pandemic sera as controls for the primary antibody (Figure 3. We observed a clear green signal in cells transfected with pcDNA3.1/D-S1, pcDNA3.1/D-S1N, and pcDNA3.1/O-SN, suggesting that the fusion proteins had regions and conformations similar to those of native SARS-CoV-2 proteins. None of the Vero cells transfected with the different plasmids showed a positive signal when pre-pandemic serum samples were used.

### 3.3. Different Doses of Naked DNA Induce a Humoral Immune Response in Mice

The dose of DNA administered to the mice was then determined. Groups of three BALB/c mice were immunized with three different doses to evaluate the immune response induction. To evaluate the response against the selected S and N regions, we used previously reported S and N proteins [31]. Six groups of three mice were used, and 10, 20, and 40 µg doses of plasmids were administered (Figure 4a). Twenty days after immunization, the mice were bled, and serum samples were obtained to evaluate IgM- and IgG-specific antibody responses against S1 and N proteins.

The IgM antibody response against the S1 protein (Figure 4b–d) was observed in the groups immunized with 20 and 40 µg doses of pcDNA3.1/S1, pcDNA3.1/D-S1, pcDNA3.1/D-S1N, and pcDNA3.1/O-SN. Furthermore, IgM antibodies against the N protein (Figure 4e–g) were detected in groups immunized with pcDNA3.1/N, pcDNA3.1/D-S1N, and pcDNA3.1/O-SN at a higher dose of 40 µg. S1 IgG was detected in pcDNA3.1/S1, pcDNA3.1/D-S1, pcDNA3.1/D-S1N, and pcDNA3.1/O-SN (Figure 4h–j). Doses of 20 and 40 µg induced the production of antibodies on day 20, which subsequently increased. In groups immunized with 10 µg, we detected S1 IgG within just 60 days. Abs against N were detected in the pcDNA3.1/N, pcDNA3.1/D-S1N, and pcDNA3.1/O-SN groups (Figure 4k–m). Similarly to S1 IgG, for groups immunized with 10 µg, we detected N IgG antibodies on day 60. For 20 and 40 µg doses, we detected N IgG on day 20, which subsequently increased.

Mice immunized with 10 µg (h, k) of plasmids showed the lowest levels of S1 and N antibodies. Mice immunized with 20 µg (i, l) or 40 µg (j, m) of plasmids showed S1 or N IgG antibodies from day 20 onwards, and a higher peak on day 60. S1 IgM and IgG were detected in mice immunized with the plasmids pcDNA3.1/S1, pcDNA3.1/D-S1, pcDNA3.1/D-S1N, and pcDNA3.1/O-SN. N IgM and IgG antibodies were detected only in mice immunized with pcDNA3.1/N, pcDNA3.1/D-S1N, and pcDNA3.1/O-SN.

### 3.4. Immunization with pcDNA3.1/D-S1, pcDNA3.1/D-S1N, and pcDNA3.1/O-SN Induce Specific Humoral Immune Responses

A new immunization schedule was performed using a dose of 25 µg of plasmids (Figure 5a). The elicited humoral immune response was evaluated using ELISA, and serial dilutions of serum samples from the immunized groups were used to determine the titers of IgM and IgG antibodies against the S1 and N proteins. We detected a higher peak of S1 IgM after 20 days, which subsequently decreased in the pcDNA3.1/S1, pcDNA3.1/D-S1, pcDNA3.1/D-S1N, and pcDNA3.1/O-SN groups (Figure 5b). A higher peak of anti-N IgM was detected 20 days after immunization and subsequently decreased in pcDNA3.1/N, pcDNA3.1/D-S1N, and pcDNA3.1/O-SN (Figure 5b). The titers of IgM antibodies against N protein were similar between the pcDNA3.1/N and pcDNA3.1/D-S1N groups. In contrast, IgG antibody titers were considerably higher than the IgM levels and increased steadily in the different groups over time. The specific anti-S1 IgG titers (Figure 5c) of pcDNA3.1/S1 and pcDNA3.1/D-S1N were similar. Interestingly, on day 40, the S1 titer of pcDNA3.1/D-S1N was significantly higher than that of pcDNA3.1/D-S1. The specific anti-N IgG titers (Figure 5f) of pcDNA3.1/N and pcDNA3.1/D-S1N were similar, and no statistically significant differences were observed. The titers of this group were significantly higher than those of pcDNA3.1/O-SN. IgM and IgG antibodies were not detected in groups immunized with the parental plasmid. Additionally, the IgG titers between days 60 and 180 in half of the mice from each group were compared (Figure 5d,g). We observed a decline in the S1 and N titers. However, these levels were higher than those in the group immunized with the parental plasmid pcDNA3.1.

It is well known that IgG1 antibodies are potent activators of ADCC and possess an affinity for all Fc receptors. In contrast, IgG2 antibodies possess the ability to neutralize viruses. Therefore, determining the proportions of these immunoglobulins is important. DNA immunization induced the production of IgG1 and IgG2 antibodies against the S1 and N proteins (Figure 5h,i). The results showed that IgG1 levels were predominantly higher in all groups.

Additionally, we evaluated the neutralizing activity of these IgG antibodies by performing an in-house surrogate virus neutralization test according to a previous report [29]. We determined the positive neutralization activity (upper 30%) in groups immunized with pcDNA3.1/S1, pcDNA3.1/D-S1, pcDNA3.1/D-S1N, and pcDNA3.1/O-SN (Figure 5j). We observed a higher level of neutralization activity in the sera of mice immunized with pcDNA3.1/D-S1N and pcDNA3.1/O-S1N. Sera from pcDNA3.1/D-S1N mice showed a significantly higher percentage of neutralization than sera from pcDNA3.1/S1 and pcDNA3.1/D-S1 mice. No statistically significant differences were observed between the pcDNA3.1/O-SN, pcDNA3.1/S1, and pcDNA3.1/D-S1 cells. Thus, we evaluated serial dilutions of mouse sera and determined the titers of neutralization activity (Figure 5k). We found that sera from pcDNA3.1/S1, pcDNA3.1/D-S1, and pcDNA3.1/O-SN-immunized mice showed neutralization activity, even at a 1:80 dilution. Sera from the pcDNA3.1/D-S1N mice, which had the highest neutralization capacity, showed neutralization activity at a 1:160 dilution.

### 3.5. DNA Immunization with pcDNA3.1/D-S1, pcDNA3.1/D-S1N, and pcDNA3.1/O-SN Induces Specific Cellular Immune Responses

Cellular immune response is crucial for the resolution of SARS-CoV-2 infection. Therefore, we analyzed this response in mice immunized with the different plasmids.

Splenocytes were obtained after the third immunization, cultured, stimulated with S1 or N protein, and analyzed using flow cytometry (FC) (Figure 6a) to analyze the production of IFN-γ by CD4^+^ cells (Figure 6b). We observed the highest levels of IFN-γ after stimulation with the N protein, particularly in the pcDNA3.1/N and pcDNA3.1/O-SN groups. IFN-γ production after S1 stimulus was higher than that which occurred without stimulus, but less than that with the N stimulus. We also analyzed the IFN-γ production in CD8^+^ cells (Figure 6c) and found that S1 stimulus induced more IFN-γ production compared to that which occurred under no-stimulus conditions. Stimulus with N also induced IFN-γ production in the pcDNA3.1/N group. Interestingly, CD8^+^ cells from the pcDNA3.1/O-SN group had higher levels of IFN-γ after S1 stimulus. Additionally, a proliferation assay was performed and the total number of CD3^+^ cells after stimulation with S1 or N proteins was analyzed (Figure 6d,e). After S1 stimulation, we observed proliferation of pcDNA3.1/S1, pcDNA3.1/D-S1, pcDNA3.1/D-S1N, and pcDNA3.1/O-SN cells. The highest proliferation was observed in the pcDNA3.1/S1 and pcDNA3.1/D-S1 groups. However, the highest proliferation among the N-induced groups was observed in pcDNA3.1/N. Notably, proliferation was not observed in the pcDNA3.1 group.

### 3.6. Antibodies Induced by DNA Immunization Recognized SARS-CoV-2 Viral Proteins in Infected Cells

After immunization, we assessed whether serum samples obtained from mice immunized with the plasmids could recognize the native proteins produced by SARS-CoV-2. This was performed using Vero cells infected with the SARS-CoV-2 isolate (this experiment was performed in a BSL-3 laboratory). A serum pool from different groups of mice after 60 days of immunization with plasmids was used as the primary antibody. A positive signal was observed when the sera from mice immunized with pcDNA3.1/S1, pcDNA3.1/N, pcDNA3.1/D-S1, pcDNA3.1/D-S1N, or pcDNA3.1/O-SN were used (Figure 7). Interestingly, we observed that a positive signal (red) from the pcDNA3.1/O-SN group had the lowest intensity compared to the signals from the other groups. Sera from mice immunized with the parental plasmid pcDNA3.1 did not react with infected cells. In addition, the pooled immune sera did not react with uninfected Vero cells.

### 3.7. DNA Immunization with pcDNA3.1/D-S1, pcDNA3.1/D-S1N, and pcDNA3.1/O-SN Elicited Cross-Reaction with RBD of SARS-CoV-2 VOCs

SARS-CoV-2 Alpha, Gamma, and Delta variants have been identified as VOCs with high transmissibility and, in the case of Delta, disease severity. To investigate whether the plasmids encoding Delta and Omicron subunits can elicit antibodies and T cell responses against different RBDs of VOCs, sera obtained from BALB/c mice 60 days after priming were tested using ELISA. FC was performed using T cells from the spleens of the immunized mice (Figure 8). We observed that sera from mice immunized with pcDNA3.1/S1 recognized the RBD protein from Alpha, Gamma, and Delta variants of concern (VOCs) (Figure 8a). These sera also recognized the XB4.5 and BQ1.1 subvariants of the Omicron strain; however, the responses were minor compared to those for other VOCs and the Wuhan RBD. Sera from mice immunized with pcDNA3.1/D-S1 and pcDNA3.1/D-S1N (Figure 8b,c) also recognized the RBD of all the VOCs evaluated. Both showed cross-reactivity with Omicron subvariants, and a decrease in OD was observed; this OD was more significant than that in the control group. As expected, sera from pcDNA3.1/O-SN mice (Figure 8d) recognized all RBD subvariants of Omicron and Alpha, Gamma, and Delta VOCs. No significant differences were observed between groups. When we analyzed IFN-γ production (Figure 8e,f) after stimulus with RBD from Delta, we observed a significant response in CD4^+^ and CD8^+^ cells from mice immunized with pcDNA3.1/S1 and pcDNA3.1/D-S1N. This response was also observed in the pcDNA3.1/D-S1N and pcDNA3.1/O-SN groups. On the other hand, after stimulation with Omicron XB4/5 RBD, we observed increased IFN-γ in CD4^+^ and CD8^+^ cells from mice immunized with pcDNA3.1/O-SN. Production of IFN-γ was observed in the pcDNA3.1/D-S1 and pcDNA3.1D-S1N groups; however, it was lower than in the pcDNA3.1/O-SN group. Finally, the percentage of proliferation was determined after stimulus with RBD from Delta or Omicron XB4/5 (Figure 8g). We observed a proliferation of approximately 10% in the pcDNA3.1/S1, pcDNA3.1/D-S1, pcDNA3.1/D-S1N, and pcDNA3.1/O-SN cells in response to the two variants. No statistically significant differences were observed between groups.

## 4. Discussion

In May 2023, the World Health Organization declared an end to COVID-19’s emergent status. New COVID-19 vaccines are necessary to deal with the SARS-CoV-2 variants of concern and develop long-lasting immunity. Thus, the next generation of COVID-19 vaccines were designed to evaluate new platforms and new antigens.

The SARS-CoV-2 N protein has attracted increasing interest as an alternative antigen that can induce a robust and long-lasting cellular immune response. Interestingly, strong humoral and cellular responses have been observed in studies that combined spike/RBD and whole nucleocapsids in immunization schemes [22,32,33,34]. Castro et al. [34] developed a fusion protein named “SpiN” that includes only some immunogenic regions from spike and nucleocapsid protein. Immunization of mice with SpiN induced an immune response and improved survival after SARS-CoV-2 challenge [35]. Thus, in this study, we designed fusion proteins with the most immunogenic regions of the SARS-CoV-2 spike and nucleocapsid proteins using bioinformatic approaches. Two of these fusion proteins were based on the Delta strain, and the other on the Omicron B.5 strain. Bioinformatics analysis suggested that using these proteins as antigens could induce an immune response.

Protein-based vaccines use adjuvants capable of activating innate receptors such as TLRs, thereby improving adaptive responses [35,36]. Molecular docking of the constructs showed that the resultant fusion proteins could bind to TLR3 and TLR4 receptors. This suggests the possibility of innate response activation. Furthermore, when Vero cells were transfected with plasmids containing pcDNA3.1/D-S1, pcDNA3.1/D-S1N, and pcDNA3.1/O-SN, the pooled sera from the infected COVID-19 cells recognized the three proteins. Analysis of extracellular expression using Western blot revealed three proteins with molecular weights that were higher than predicted, which could be due to post-transcriptional changes such as glycosylation. In our bioinformatics analysis, we identified possible glycosylation sites that have been reported in spike and nucleocapsid proteins [37,38].

Our study highlights the induction of an efficient humoral response by intramuscular immunization with naked plasmids at doses of 20–40 µg, similar to that observed by Wang et al. [39]. They reported that doses of 2.5 to 40 µg of naked pcDNA3.1/IgE-spike-S1/S2-D614G-6P-foldo administered through electroporation induced S IgG antibodies. Most DNA immunization schemes use formulations such as liposomes and delivery systems such as electroporation [40,41].

We decided to use a dose of 25 µg for subsequent immunization schemes. The titers of N IgG antibodies were similar in the pcDNA3.1/N and pcDNA3.1/D-S1N groups. Notably, the titers induced by pcDNA3.1/O-SN were the lowest. However, most of the N regions used in O-SN were selected to induce a cellular response; therefore, the lowest titers were observed. Babuadze et al. reported the induction of spike antibodies after immunization with 50 µg of naked DNA [42]. Our immunizations using 25 µg of naked DNA resulted in efficient results, considering that we did not use electroporation. Furthermore, DNA immunization induced the production of IgG1 and IgG2 antibodies, similar to the data reported for SARS-CoV-2 antigens using recombinant proteins [22,32] or viral vectors [43]. Interestingly, our DNA immunization induced antibodies that could be detected until 4 months after the completion of the scheme. This result was similar to that reported in humans after SARS-CoV-2 infection or vaccine immunization [44,45].

Neutralizing antibody activity is essential for resolving SARS-CoV-2 infections [46]. In our study, we used an in-house surrogate virus neutralization test to determine the neutralizing capacity of antibodies present in the sera of immunized mice to block the interaction between the RBD and ACE2 [28]. Among the different immunization groups, sera from mice immunized with pcDNA3.1/D-S1N and pcDNA3.1O-SN showed the highest neutralization activity. These data support the idea that combining a spike with a nucleocapsid increases the humoral response compared with using a spike alone. Statistically significant differences were found between pcDNA3.1/S1, pcDNA3.1/D-S1, and pcDNA3.1/D-S1N but not between pcDNA3.1/S1, pcDNA3.1/D-S1, and pcDNA3.1/O-SN.

Relevant data from immunofluorescence assays indicate that antibodies induced by DNA immunization can recognize SARS-CoV-2 in infected cells. Moreover, we observed similar fluorescence signals in the sera from pcDNA3.1/S1-, pcDNA3.1/N-, pcDNA3.1/D-S1-, and pcDNA3.1/D-S1N-immunized mice; however, sera from animals immunized with pcDNA3.1/O-SN had the weakest signal. Because the cells used in these experiments were originally infected with Wuhan SARS-CoV-2 and the sera were from animals immunized with plasmids based on the Omicron strain, we assumed that some epitopes were missing; therefore, a diminished signal was observed. These findings are important because they suggest that immunized mice may respond to native SARS-CoV-2 antigens upon infection.

In a study by Guimaraes et al. [47], using lipid nanoparticle-encapsulated plasmids conferred protection in vaccinated animals against SARS-CoV-2. In contrast, we observed IgG Ab, neutralization activity, and virus recognition, and expected that the immune response induced by naked DNA immunization could resolve SARS-CoV-2 infection.

The cellular immune response induced by the nucleocapsid helps resolve the SARS-CoV-2 infection [19]. To analyze this response after immunization with our plasmids, we determined levels of CD4^+^ and CD8^+^ T cells that were positive for IFN-γ. We detected mainly CD4^+^/IFN-γ cells after the stimulus with N, and mainly detected CD8^+^/IFN-γ in response to the S1 stimulus. These findings correspond to those reported in the literature, where N mainly activates CD4^+^ T cells and S mainly activates CD8^+^ T cells [48]. Dangi et al. [21] reported higher levels of IFN-γ in mice immunized with Ad5-N+S compared to those immunized with Ad5-S or Ad5-N. Hajnik et al. [49] reported higher levels of IFN-γ, TNF-α, and IL-2 in hamsters immunized with mRNA S+N than those immunized with mRNA S. Also, previously, we reported the highest levels of IFN-γ, TNF-α, and IL-2 in mice immunized with RBD+N and S1+N compared to immunizations just with S1 or RBD [22]. This study found higher levels of IFN-γ in mice immunized with pcDNA3.1/D-S1N and pcDNA3.1/O-SN compared to those immunized with pcDNA3.1/D-S1. Furthermore, the positive data from the proliferation assays suggested that DNA immunization generated specific T cells against the N and S proteins.

Different platforms have been used to evaluate the combinations of S and N proteins that induce immune responses against VOCs. Interestingly, many of these candidates have antigens based on the Wuhan strain and have even responded to and are protected against VOCs such as Alpha, Beta, Gamma, Delta, and Omicron [23,34,49,50,51,52,53]. However, vaccines based on sequences of different VOCs are still under development. Wussow et al. [23] explored the use of a multi-antigen synthetic modified vaccinia Ankara (MVA) vector that expressed spike and nucleocapsid proteins based on the Omicron BA.1 or Beta (B.1.351) sequence. Immunization of Syrian hamsters with MVA-N/S Beta (COH04S351) and Omicron (COH04S529) induced neutralization activity against different Omicron subvariants and protected these animals after viral challenge [23]. In this study, we designed two fusion proteins based on Delta and Omicron SARS-CoV-2 strains. Similarly to results reported in previous studies, cross-reactivity was observed. Humoral and cellular cross-reactions were mainly observed in T cells from the pcDNA3.1/O-S1N group, whereas the latter were observed in T cells from the pcDNA3.1/D-S1N group. These findings support the cross-protection observed in vaccinated populations that are partially protected against VOCs. This also supports the strategy of incorporating the sequences of circulating variants and new vaccines that can be used as booster doses [54]. Another limitation of our study is that it had few RBD variants and could not evaluate a more complete cellular response.

One limitation of our study is that we used simulated neutralization tests to assay neutralization and not challenge experiments with SARS-CoV-2 because we had limited access to a BSL Level 3 facility.

## 5. Conclusions

Immunization with naked DNA-encoding fusion proteins of SARS-CoV-2 induced humoral and cellular responses. Additionally, this response exhibited a cross-reaction with Delta and Omicron VOCs.

## Figures and Tables

**Figure 1 vaccines-13-00134-f001:**
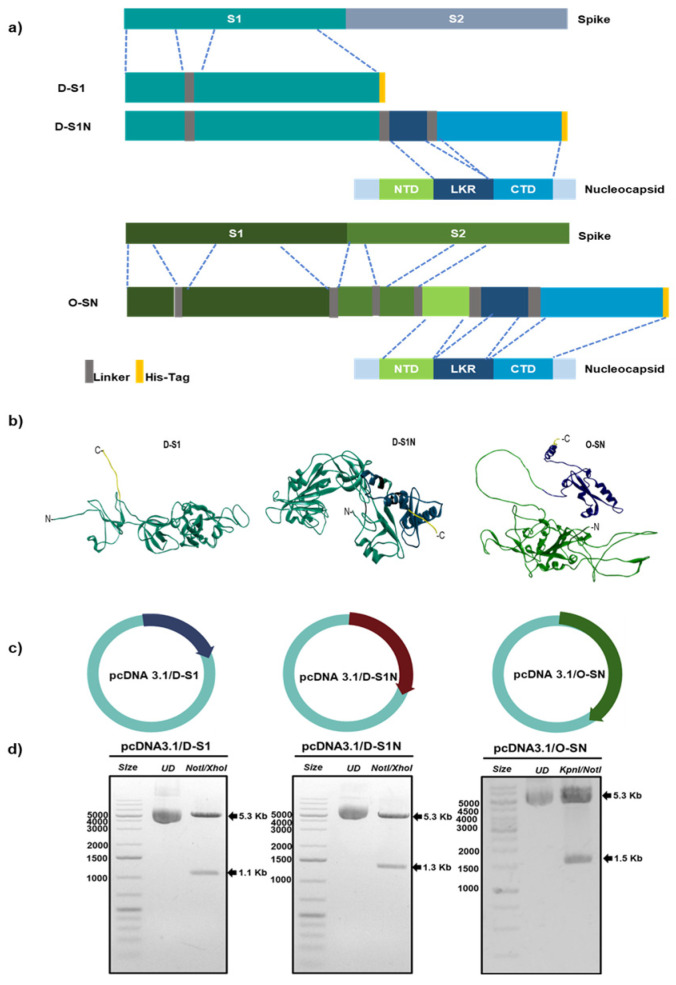
Design and cloning of pcDNA3.1/D-S1, pcDNA3.1/D-S1N, and pcDNA3.1/O-SN. (**a**) Representation of immunogenic regions of the spike and nucleocapsid proteins used in the design of the fusion proteins. (**b**) Structures of D-S1, D-S1N, and O-SN predicted by Alpha Fold. (**c**) Scheme of D-S1, D-S1N, and O-SN sequence cloning into plasmid pcDNA3.1. (**d**) Enzymatic digestion of synthesized plasmids.

**Figure 2 vaccines-13-00134-f002:**
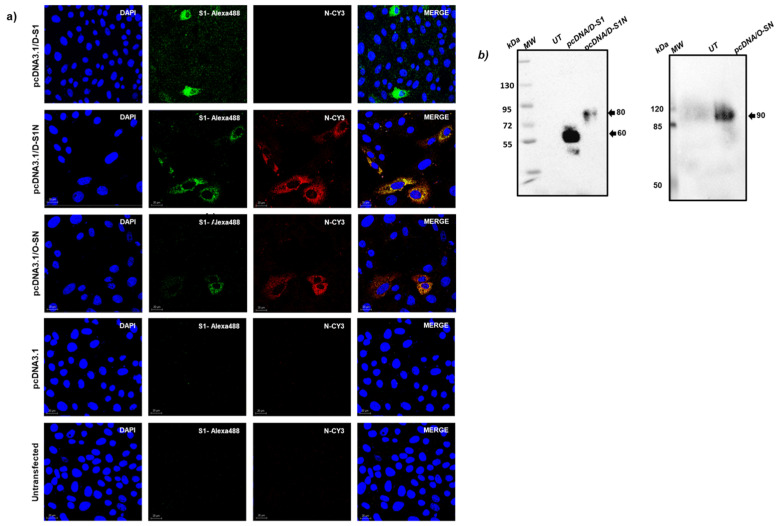
Intracellular and extracellular expression of fusion proteins D-S1, D-S1N, and OS-N. (**a**) Vero cells were transfected with pcDNA3.1/D-S1, pcDNA3.1/D-S1N, or pcDNA3.1/O-S1N, and 24 h after transfection, antibodies against spike and nucleocapsid were used. s. (**b**) Extracellular expression was evaluated in supernatants from Expi293 cells 5 days after transfection by Western blot using anti-His antibody.

**Figure 3 vaccines-13-00134-f003:**
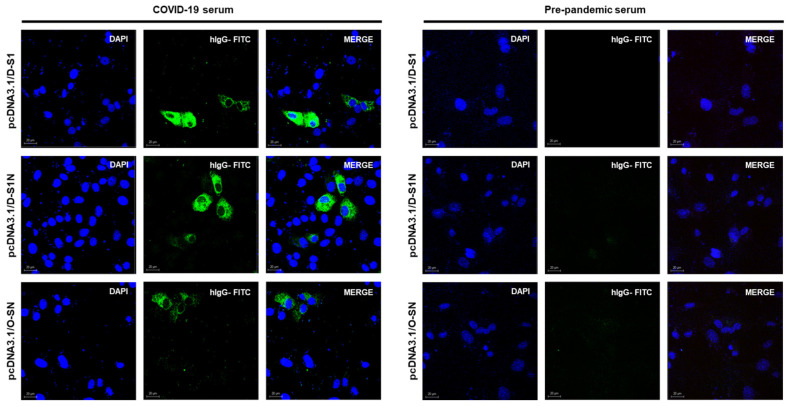
Specific recognition of fusion proteins using serum samples from patients with COVID-19. Vero cells were transfected with pcDNA3.1/D-S1, pcDNA3.1/D-S1N, or pcDNA3.1/O-SN. The recognition of fusion proteins was evaluated 24 h post transfection by an immunofluorescence assay after treatment with human serum samples from patients with COVID-19.

**Figure 4 vaccines-13-00134-f004:**
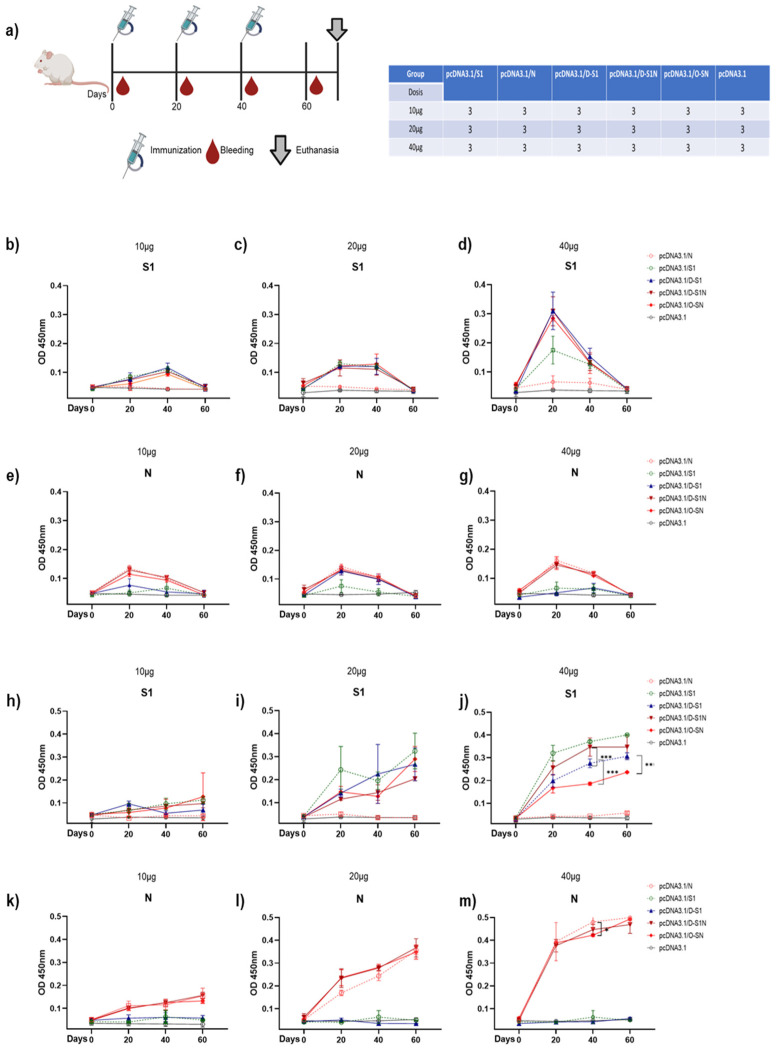
Selection of DNA doses to induce an immune response in BALB/c mice. (**a**) Scheme of immunization with plasmids in BALB/c mice. Six groups were immunized with doses of 10 µg, 20 µg, or 40 µg of plasmids. The humoral response was measured using an enzyme-linked immunosorbent assay (ELISA). Measurement of IgM antibodies against S1 (**b**–**d**) and N (**e**–**g**), and IgG antibodies against S1 (**h**–**j**) and N (**k**–**m**). Bars represent the mean ± SD. * *p* < 0.05, ** *p* < 0.01, *** *p* < 0.001. *p* ≤ 0.05 was considered statistically significant.

**Figure 5 vaccines-13-00134-f005:**
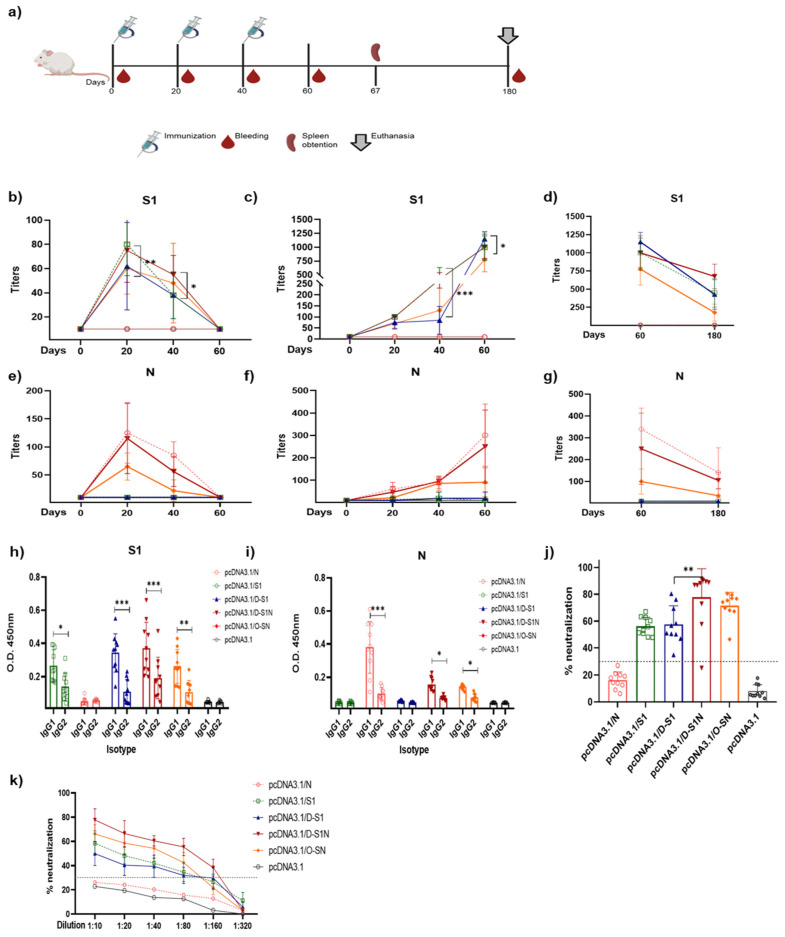
Humoral immune response induced by immunization with pcDNA3.1/D-S1, pcDNA3.1/D-S1N, and pcDNA3.1/O-SN. (**a**) Scheme of immunization with plasmids in BALB/c mice. Ten mice per group were immunized with 25 µg of plasmids. IgM and IgG antibody titers were determined using ELISA. S1 IgM (**b**) antibodies. S1 IgG antibodies were detected in the same groups (**c**). N IgM (**e**) antibodies were observed in pcDNA3.1/N, pcDNA3.1/D-S1N, and pcDNA3.1/O-SN groups. N IgG antibodies were detected in the same groups (**f**), and the highest titers of antibodies were observed on day 60. To determine the duration of S1 and N IgG antibodies, half of the mice were bled on day 180. Comparative analysis of titers on 60 and 180 days was performed (**d**,**g**). Sera obtained on day 60 was used to determine S1 and N IgG1/IgG2 levels (**h**,**i**). Antibodies with neutralization activity were observed (**j**,**k**). (**j**) Serial dilution of sera was used to determine neutralization activity (**k**). Bars represent the mean ± SD. * *p* < 0.05, ** *p* < 0.01, *** *p* < 0.001. *p* ≤ 0.05 was considered statistically significant.

**Figure 6 vaccines-13-00134-f006:**
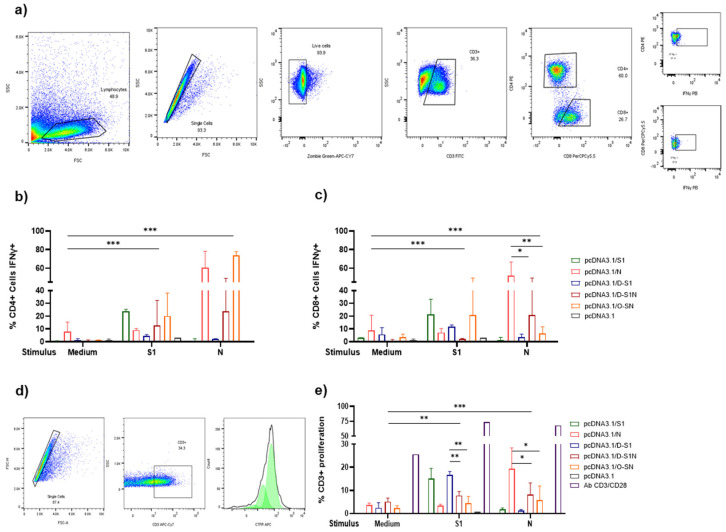
Cellular immune response induced by immunization with pcDNA3.1/D-S1, pcDNA3.1/D-S1N, and pcDNA3.1/O-SN. IFN-γ levels in CD4^+^ and CD8^+^ T cells were determined by flow cytometry. (**a**) The percentage of CD4^+^ cells positive for IFN-γ was determined before (NE) and after the stimulus with S1 or N proteins. (**b**) The percentage of CD8^+^ cells positive for IFN-γ was determined before and after the stimulus with S1 or N proteins. (**c**) The proliferation in CD3^+^ cells were determined 5 days after culture and stimulus with S1 or N protein (**d**,**e**). Bars represent the mean ± SD. * *p* < 0.05, ** *p* < 0.01, *** *p* < 0.001. *p* ≤ 0.05 was considered statistically significant.

**Figure 7 vaccines-13-00134-f007:**
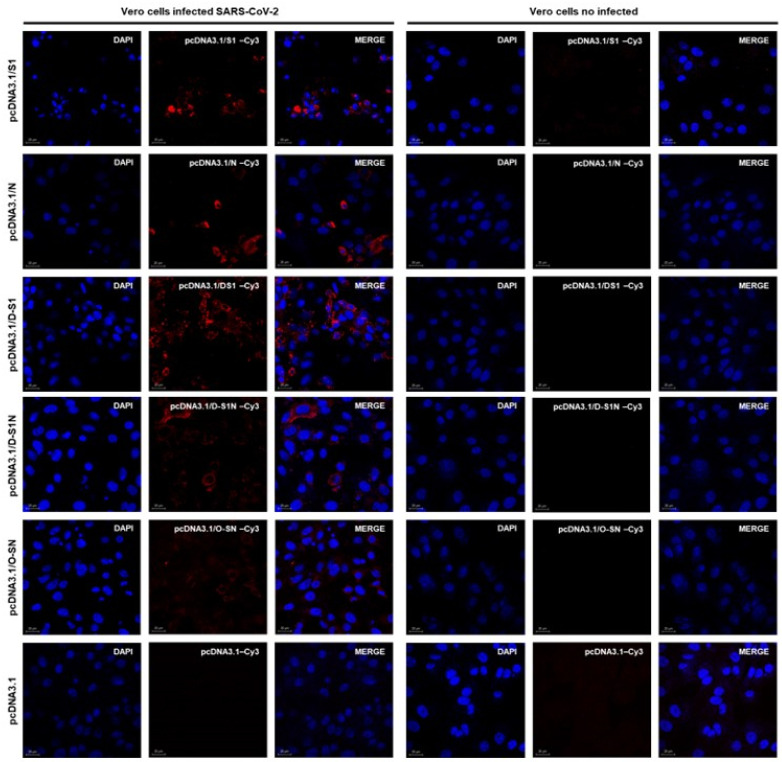
SARS-CoV-2 is recognized by sera from plasmid-immunized mice. Vero cells were infected with SARS-CoV-2 for 24 h and were exposed to pooled hyperimmune sera from immunized mice. Positive red signals were recognized in SARS-CoV-2 found within sera from pcDNA3.1/S1, pcDNA3.1/N, pcDNA3.1/D-S1, pcDNA3.1/D-S1N, pcDNA3.1/O-SN immunized mice.

**Figure 8 vaccines-13-00134-f008:**
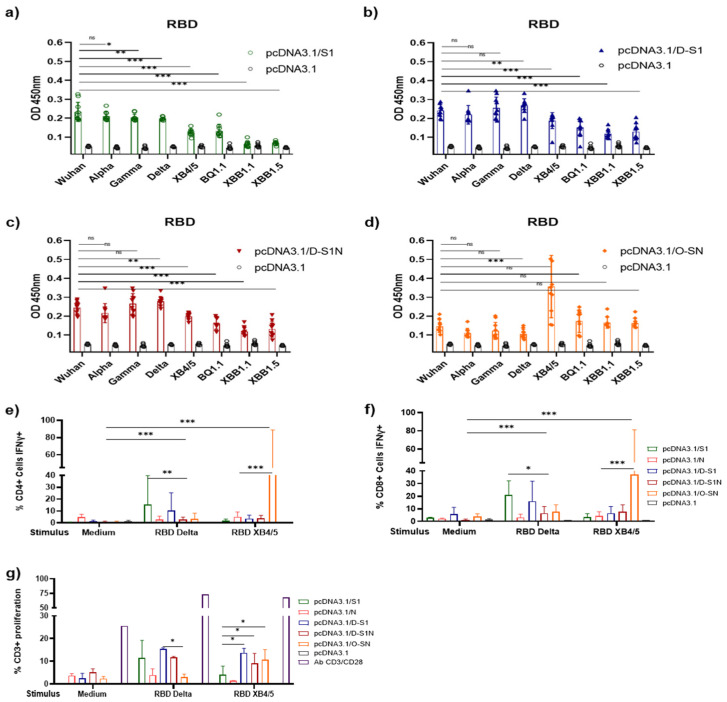
Immune response against VOCs. (**a**–**d**) Humoral cross-reaction was determined using ELISA, where the plaques were coated with RBD proteins purified from Wuhan, Alpha, Gamma Delta BQ 1.1, XBB.1, and XBB1.5 variants of concern. Immune responses were observed using serum from the groups immunized with (**a**) pcDNA3.1/S1, (**b**) pcDNA3.1/D-S1, (**c**) pcDNA3.1/DS1N, and (**d**) pcDNA3.1/O S1N. (**e**–**g**) Cellular response against Delta and Omicron XB4/5 RBD. The percentage of CD4^+^ cells positive for IFN-γ (**e**). The percentage of CD8^+^ cells positive for IFN-γ (**f**). Proliferation after stimulus with the two RBD variants (**g**). Bars represent the mean ± SD. * *p* < 0.05, ** *p* < 0.01, *** *p* < 0.001. *p* ≤ 0.05 was considered statistically significant. ns = no significant.

## Data Availability

The data presented in this study are available upon request from the corresponding author.

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
