# Peer review of "Vaccination with Plasmids Encoding the Fusion Proteins D-S1, D-S1N and O-SN from SARS-CoV-2 Induces an Effective Humoral and Cellular Immune Response in Mice"

_vaccines, 2025, doi:10.3390/vaccines13020134_

Round 1

Reviewer 1 Report

Comments and Suggestions for Authors

The current manuscript reported the development and evaluation of DNA vaccines encoding fusion proteins combining immunogenic regions of SARS-CoV-2 spike and nucleocapsid proteins. By using in silicon analysis, three constructs (D-S1, D-S1N, and O-SN) were designed based on Delta and Omicron variants. The authors proved that these constructs can efficiently express in cells and elicit strong humoral and cellular immune response in mice. The induction of innate antiviral activity by these DNA constructs were also determined. The cross-reactivity of the induced immunogenicity against other SARS-COV-2 variants were also determined. The experiments were well designed and carefully conducted, followed by proper data analysis,interpretation and discussion. Although, it lacks a viral challenge study to eventually determine the protective efficacy of the reported DNA vaccine constructs, the presented data shows that these constructs are promising candidates for future vaccine development. I only have few points to discuss about.

Major points:

1, What is the rationale behind the development of the constructs based on Delta and Omicron B.5?

2. Any deleterious effect on the mice after immunizations of the reported DNA vaccine constructs?

3. Although the addition of the nucleocapsid regions enhanced the immune responses, the authors mentioned little about why this would be the case in the discussion part.

Minor points:

1.     Lack information of the figure legends in Figure. It is better to add extra information about what does the different color (channel) stand for, and how many days after transfection were the samples tested in the figure 2, 3 and 7. Besides, it needs more in details of the statistics used in the images.

2.     Figure 3. One of images on the right panel (in the middle) is smaller than the others.

3.     Figure 4. There are two lines of images e-g and k-n that are labeled as measurement by antibodies against N.

Author Response

Reviewer 1

Major points:                                                                            

1.- What is the rationale behind the development of the constructs based on Delta and Omicron B.5?

It is well known that the original virus that started the SARS-Cov2 pandemic was the Wuhan strain. However, as the time pass, variants have emerged that have spread throughout the world. Among which, in 2021 in Mexico Delta strains were present, and therefore we decided star with this variant. Furthermore, in 2022 Omicron B.5 was one of the variants that have high prevalence in the world therefore, we select this other sequence for our third vaccine candidate.

2.- Any deleterious effect on the mice after immunizations of the reported DNA vaccine constructs?

We carried out a detailed record of any effects that we could observe throughout the experiments. However, we did not observe any effects adverse in immunized mice.

3.- Although the addition of the nucleocapsid regions enhanced the immune responses, the authors mentioned little about why this would be the case in the discussion part.

As we described in the paper, there are several reports that strongly suggest that the N protein, improves the immune response trough the induction of an effective cellular immune response mediated by CD4 cells. However, to date no conclusive mechanism for this phenomenon has been described.

In our paper, we also report that combination with N induces production IL-2, cytokine import in the differentiation and maturation of B cells to plasmatic cells. Additionally, in other studies that evaluated the combination of spike with nucleocapsid, they found a better immune response in the combination in comparison with immunization with only spike.

Minor points:

1.-Lack information of the figure legends in Figure. It is better to add extra information about what does the different color (channel) stand for, and how many days after transfection were the samples tested in the figure 2, 3 and 7. Besides, it needs more in details of the statistics used in the images.

Hours of post-transfection and infection have been added in legends of figures.

  1. Figure 3. One of images on the right panel (in the middle) is smaller than the others.

The figure has been changed

  1. Figure 4. There are two lines of images e-g and k-n that are labeled as measurement by antibodies against N.

Is correct, e-g (e, f and g) are IgM antibodies against N. And k-M (k, l, M) are IgG antibodies against N

Reviewer 2 Report

Comments and Suggestions for Authors

Does the 8His tag have any clinical effect on the mice?

Excellent, well thought out paper.

How would 25 ug of plasmid DNA used for mouse inoculation translate into humans? Would this be more or less expensive than RNA vaccines?

Is there a way to determine how much of the naked DNA was degraded during immunization? Do you think this strategy could be used for other pathogens? If so, which ones?

Author Response

Reviewer 2

1.- Does the 8His tag have any clinical effect on the mice?

So far, no clinical effects on mice immunized with proteins with His-tags have been reported. Furthermore, we carried out a detailed record of any effects that we could observe throughout the experiments. However, we did not observe any effects adverse in immunized mice Also, no clinical effects were shown in our vaccinated mice.

2.- How would 25 ug of plasmid DNA used for mouse inoculation translate into humans? Would this be more or less expensive than RNA vaccines?

It is difficult yet to translate our vaccines to humans. At this point, we are presenting only a preclinical assay, evaluating the capacity of naked immunization with these plasmids to induce an immune response. The next step is to improve the immune response and reduce the DNA dose by coupling the plasmid with carriers such as liposomes or another nanoparticle. After this point, it could be possible to escalate the doses for bigger models (primates and even humans).  With a lower dose, DNA vaccines could be cheaper than mRNA vaccines. Because the yield in production is higher than RNA, and also, DNA vaccines are easy to store in comparison with RNA vaccines.

3.-Is there a way to determine how much of the naked DNA was degraded during immunization?

Is not possible to determinate the amount of DNA that is degraded during the immunization. Thus, there are some work focused on increase or protect the DNA to the proteases during the immunization.

3.- Do you think this strategy could be used for other pathogens? If so, which ones?

The DNA vaccination is a strategy well described for other pathogens. Exhibing significant advantages over traditional vaccines regarding their ability to induce CD4+ and CD8+ T cell responses. Our group has worked with this strategy since some years ago using sequences of FMDV (doi: 10.1099/0022-1317-82-7-1713). also with DENV(doi: 10.4161/hv.25673). Interestingly clinical assays has been already performed by other groups as example of this, the vaccines called VGX-3100 (NCT01304524) and the GX-188E vaccine (NCT01634503) both encoding immunogenic peptides based on E6 and E7 genes of HPV-16 and 18, in patients with promising results (Lancet 2015, 386, 2078–2088). Furthermore in India a plasmid vaccines encoding SARS-CoV-2 call ZyCOV-D vaccine proteins has been used in humans in phase II/III trials (doi: 10.1016/S0140-6736(22)00151-9